# Mycotoxin DON Accumulation in Wheat Grains Caused by Fusarium Head Blight Are Significantly Subjected to Inoculation Methods

**DOI:** 10.3390/toxins14060409

**Published:** 2022-06-15

**Authors:** Limei Xian, Yuhui Zhang, Yi Hu, Suqin Zhu, Zhuo Wen, Chen Hua, Lei Li, Zhengxi Sun, Tao Li

**Affiliations:** Key Laboratory of Plant Functional Genomics of the Ministry of Education/Jiangsu Key Laboratory of Crop Genomics and Molecular Breeding/Collaborative Innovation of Modern Crops and Food Crops in Jiangsu/Jiangsu Key Laboratory of Crop Genetics and Physiology, College of Agriculture, Yangzhou University, Yangzhou 225009, China; limeixian0710@foxmail.com (L.X.); yuhuizhang123@foxmail.com (Y.Z.); yihu0221@foxmail.com (Y.H.); zhuqqfc@foxmail.com (S.Z.); skiumy@foxmail.com (Z.W.); chenhua0729@foxmail.com (C.H.); lilei@yzu.edu.cn (L.L.); zhengxisun@yzu.edu.cn (Z.S.)

**Keywords:** wheat, fusarium head blight (FHB), inoculation methods, deoxynivalenol (DON), disease severity, premature spike death (PSD)

## Abstract

The disease severity and mycotoxin DON content in grains caused by fusarium head blight (FHB) have been two prioritized economical traits in wheat. Reliable phenotyping is a prerequisite for genetically improving wheat resistances to these two traits. In this study, three inoculation methods: upper bilateral floret injection (UBFI), basal bilateral floret injection (BBFI), and basal rachis internode injection (BRII), were applied in a panel of 22 near-isogenic lines (NILs) contrasting in *Fhb1* alleles. The results showed that inoculation methods had significant influence on both disease severity and mycotoxin accumulation in grains, and the relationship between them. UBFI method caused chronic FHB symptom characterized as slow progress of the pathogen downward from the inoculation site, which minimized the difference in disease severity of the NILs, but, unexpectedly, maximized the difference in DON content between them. The BBFI method usually caused an acute FHB symptom in susceptible lines characterized as premature spike death (PSD), which maximized the difference in disease severity, but minimized the difference in DON content in grains between resistant and susceptible lines. The BRII method occasionally caused acute FHB symptoms for susceptible lines and had relatively balanced characteristics of disease severity and DON content in grains. Therefore, two or more inoculation methods are recommended for precise and reliable evaluation of the overall resistance to FHB, including resistances to both disease spread within a spike and DON accumulation in grains.

## 1. Introduction

Fusarium head blight (FHB) is a worldwide fungal disease mainly caused by *Fusarium graminearum* species complex and has been reported in America [1,2], Asia [3,4], Europe [5], Australia [6], and some other countries. When favorable conditions, such as optimal temperature and humidity, occur during the flowering period of wheat, the ascospores of fusarium in soil, stubble, and other carriers will spread with rain and wind, and then colonize and infect wheat florets, and eventually affect the grain filling of wheat spikes, which has a serious impact on crop yield and food security [7]. In addition, the mycelial tip secretes the trichothecene toxins [8], which are virulent factors that aggravate the occurrence of FHB [9]. Trichothecenes, such as deoxynivalenol (DON), nivalenol (NIV), T-2 toxin, etc., accumulate in infected cereals and contaminate agricultural products, threatening the health of humans and animals [10]. Development of resistant varieties is the most effective and environmentally safe approach to manage FHB. The resistant types are generally classified into five categories: resistance to initial infection (type I) [11], disease spread from one spikelet to another (type II) [11], mycotoxin accumulation (type III) [12], kernel infection (type IV), and tolerance to yield loss (type V) [13]. Type IV and type V can be merged because both actually reflect the disease resistance of grains [14]. Resistances to FHB are typical quantitative traits controlled by one to three major quantitative trait loci (QTL) and several minor ones, and also vulnerable to environmental factors [15]. More than 600 QTL have been detected on all 21 chromosomes of wheat [16,17]. Among them, *Fhb1* on the short arm of chromosome 3B has been considered the most stable QTL with the greatest effect on type II resistance [18]. Nevertheless, functional elucidation of this QTL still remains a mystery and controversy and requires further investigation.

Successful crop improvement is based on both genetic variation of the target trait and reliable phenotyping method [19], and the latter is also essential for fine mapping and functional analysis of QTL for FHB resistance. In order to obtain uniform and effective disease pressure, resistance testing has been basically accomplished by artificial inoculations, mainly including single-floret inoculation (SFI) and spray/grain-spawn inoculation [20]. The former is the most commonly used method to measure type II resistance by percentage of symptomatic spikelets (PSS) [21]; the latter simulates the natural incidence of FHB and is generally used to reflect type I resistance or the mixed effect of type I and II [22]. Artificial inoculations still have some uncontrollable factors, leading to poor reproducibility and consistency of the phenotypes, including the amount of inoculum inoculated, the pathogenicity and virulence of strains, inoculation methods, etc. [23]. For inoculation methods, it is hard to say which one is absolutely perfect, and different methods have certain merits and limitations according to the characteristics of FHB resistance types. Floret inoculation is able to ensure more control, such as the exact inoculation time, and its application in type II resistance is well defined and illustrated [24]; however, floret inoculation is inefficient in dealing with large-scale materials. Spray inoculation saves time and labor, and has advantages in mass selection in most breeding programs [24], but it needs to be equipped with moisturizing measures, and also lack of phenotypic precision. Basal rachis internode injection (BRII) is a relatively new inoculation method, which is implemented by injecting inoculum into the basal internode of a rachis [25]. Under the BRII method, the spreading path of fungi is from the rachis internode to spikelet through the rachis node. Thus, its advantages over the prevailing methods described above lie in exemption from the moisture-maintaining system, and the trait tends to segregate in a qualitative way [25].

Excessive production of mycotoxins is a tremendous hazard during the epidemic outbreak of FHB, and decreasing mycotoxin contamination has become one of the targets for FHB resistance breeding. Mycotoxins are the pathogenic factors in the process of pathogen infecting the host. *TRI5* mutant strain, which does not produce trichothecenes, cannot enter the rachis through infected spikelet to spread into another spikelet [26]. A positive and significant correlation between mycotoxin content and visual disease severity has been frequently reported [27,28], but weak or no correlation between them was also reported [29]. This controversial situation may be attributed to the complex genetic background of mycotoxin accumulation, sensitiveness to environmental influences, inoculation methods and other factors. The mycotoxin test is usually implemented after the occurrence or/and severity assessment of FHB being completed [23], so point inoculation and spray/spawn inoculation have been involved in phenotypic evaluation of type III resistance in practice. However, the influences of different inoculation methods or infection points on mycotoxin accumulation are currently unknown.

This study focused on type II and III resistance using three relatively novel inoculation methods including BBFI [30], BRII [25], and UBFI. BBFI and UBFI were modified from SFI method. BBFI was performed by injecting inoculum into the bilateral florets of the 5th spikelet from the bottom of a spike [30]; and UBFI was performed by injecting inoculum into the bilateral florets of the 5th spikelet in the upper part of a spike. The effect of *Fhb1* on reducing severity has been well acknowledged, whereas its effect on toxin accumulation disagreed among reports [31,32] and no convincing explanation has been acknowledged. BBFI and BRII are able to cause premature spike death (PSD) in susceptible wheat genotypes. Different from the BBFI and BRII methods, the UBFI method allows pathogens to spread downward from the inoculation point and the spikelets unreached by pathogen keep normal growth and development, thus, the PSD symptom is avoided. In this study, a panel of 22 near-isogenic lines contrasting in *Fhb1* alleles with relatively simple genetic background and relatively stable resistance level was used to compare the consequences of the three methods in evaluating disease severity and DON content in grains, advantages and disadvantages of each method, and also to re-explore the relationship between disease severity and DON content in grains. We hope the outcome of this study would be useful for further improving the accuracy and reliability of FHB phenotyping, and helpful for untangling the uncertain relationship between FHB severity and DON accumulation.

## 2. Results

### 2.1. Phenotypic Variation of PSS under the Three Methods

The basic statistics and distributions of PSS under the three methods were summarized in Appendix A, respectively. For bilateral floret inoculation, PSS of NILs showed a bias distribution in 2020 (UBFI, 0.2–0.4; BBFI, 0.8–1.0), whereas they were mainly between 0.5 and 0.9 in 2021; under the BRII method, more lines were distributed between 0.0 and 0.2 in 2021, and even with a lower mean PSS in 2020 (Appendix A). Correlations of PSS between the two years were significant (*p* < 0.05) under UBFI method and highly significant (*p* < 0.01) across the BBFI and BRII methods.

*Fhb1^+^* group and *Fhb1^−^* groups significantly differed (*p* < 0.01) in PSS across the three methods in each season (Figure 1 and Figure 2). Smaller variation in PSS among the NILs and narrower PSS gap between the two groups were observed under the UBFI method (Figure 2). In comparison, larger variation of PSS among the lines and wider difference in PSS between the two groups were observed under BBFI (Figure 2), and PSS values of the lines without *Fhb1* were close to or equal to 1.0 (Table 1). Compared with the two bilateral floret inoculation methods, the variation of PSS was smaller under the BRII method, ranging from 0.0 to 0.5 (Figure 2). The rachis of the inoculated spikes of the lines carrying *Fhb1* had an obvious bleaching phenomenon, but visible symptomatic spikelets were not/seldom observed, with PSS being close to or equal to 0.0 (Table 1 and Appendix A), significantly differing from those of the lines without *Fhb1* (Figure 2), suggesting that the pathogen was easier to spread from rachis into spikelet in susceptible lines than in the resistant lines. Highly significant correlations in PSS were observed among the three methods (Figure 3), and the BLUP values over the two seasons for each method showed the order of BBFI > BRII > UBFI in PSS gap between *Fhb1^−^* and *Fhb1^+^* groups (Table 1), suggesting the BBFI and BRII methods were advantageous over UBFI in measuring FHB severity.

### 2.2. UBFI Method Maximized DON Accumulation in Grains

Among the three methods, the maximum, mean and range of DON content under UBFI method were consistently the largest across the two years (Appendix A), and the mean DON content exceeded 1000 μg∙kg^−1^ (Appendix A). In contrast, mean DON content of the NILs under BRII method was lower than those of the other two methods in each year (Appendix A). A smaller variation of DON content was observed in 2021 than that in 2020 (Appendix A), but the mean value in 2021 was higher (Appendix A). Correlations of DON content in grains between the two years were significant (*p* < 0.05) only under UBFI method, but not significant under the other two methods (*p* > 0.05).

The average DON accumulation varied from the inoculation methods with a trend of UBFI > BBFI > BRII (Appendix A). DON content of most lines without *Fhb1* exceeded 1000 μg∙kg^−1^ under the UBFI method (Figure 2), with the highest reaching 4224.38 μg∙kg^−1^ (Appendix A). Highly significant differences (*p* < 0.01) in DON content between the groups contrasting in *Fhb1* alleles under the UBFI method and significant differences (*p* < 0.05) under the BRII method were observed, respectively, but there was no significant difference (*p* > 0.05) under the BBFI method (Figure 2). The gap of DON content between *Fhb1^−^* and *Fhb1^+^* groups under the three methods followed the order of UBFI > BRII > BBFI, with intergroup gap of UBFI being about 10 times more than that of BRII (Table 1). A significant correlation in DON content (*p* < 0.05) was also observed between the UBFI and BRII methods, but neither of the two methods correlated with BBFI in DON content (Figure 3). All these results suggested that UBFI was advantageous over the BBFI and BRII methods in assaying DON accumulation potential.

### 2.3. Premature Spike Death Could Be an Alternative Method for Measuring Disease Severity

The BBFI method easily caused premature death of the inoculated wheat spikes in the lines without *Fhb1* (Figure 1B), and the BRII method was next to BBFI in PPSD. PSD phenomenon was not observed under the UBFI method since the pathogen progressed chronically downward from the point of inoculation (Figure 1A and Appendix A). The extremum, mean, and distributions of PPSD for NILs under BBFI and BRII are shown in Appendix A. PPSD under the BBFI method was mainly distributed between 0.5 and 1 over two years; PPSD under the BRII method ranged from 0 to 0.5 in 2020, and even less PSD occurred in 2021 (Appendix A). The correlation coefficient of PPSD between the two years was significant (*p* < 0.01) only under the BBFI method.

One-way ANOVA analysis of PPSD showed that there was a highly significant difference (*p* < 0.01) between the two groups with contrasting *Fhb1* alleles across the two years under the BBFI method, and the lines without *Fhb1* were close to or equal to 1.0 (Figure 4 and Table 1). PSD also occurred in some lines with *Fhb1*, but significantly less than those lines without *Fhb1*. PPSD under the BRII method was significant at *p* < 0.01 between the two groups in 2020, but not significant in 2021 (Figure 4). PPSD of the lines carrying *Fhb1* were close to 0.0 under the BRII method (Figure 4 and Table 1), indicating that PSD under the BRII method tended to segregate in a qualitative manner. The BLUP values of PPSD over the two years showed a significant positive correlation between the BBFI and BRII methods (*p* < 0.05) (Figure 3), and BBFI caused more prematurely dead spikes than BRII did with a differential BLUP value up to 0.49 between the *Fhb1^−^* and *Fhb1^+^* groups (Table 1). These results indicated that PPSD under BBFI could be an alternative index for phenotypic assessment of FHB severity and an optimal index to assay the genetic effect of *Fhb1*.

### 2.4. Relationships among PSS, DON Content in Grains and PPSD

The relationships among PSS, DON content, and PPSD across two years are shown in Figure 3. PSS under the three methods showed significantly positive correlations with DON content under the UBFI method (*p* < 0.05), but had no, or weak, correlations with DON content under the BBFI and BRII methods. There was an expectedly significant correlation between PPSD and PSS under the BBFI method (*p* < 0.05). No significant correlations were found between PPSD and DON content under the BBFI and BRII methods, but PPSD of BBFI and BRII had significantly positive correlations with DON content under the UBFI method (*p* < 0.05). The BLUP results also suggested that correlations among PSS, DON content in grains and PPSD depended on inoculation methods (Figure 3C).

## 3. Discussion

Limitations on precise phenotyping have been perceived as a constraint to both genetic study and breeding efforts [33]. A reliable and suitable phenotyping method is a prerequisite for understanding the function and the mechanism of a gene, evaluation of germplasm, breeding cultivars, and for genetic study of economically important traits including wheat FHB resistances to disease spread (type II) and DON accumulation (type III). In this study, three different inoculation methods (UBFI, BBFI, and BRII) were applied to understand their relative advantages and limitations in assessment of disease severity, DON accumulation potential in grains, and their relationships. We found that no method was versatile for both traits.

### 3.1. BBFI Was Suitable for Assessing FHB Severity Rather Than for DON Assay

Single-floret inoculation (SFI) has been widely used for resistance evaluation of FHB [20]. SFI has high labor cost and low inoculation throughput, and it is relatively stable and becomes a popular method for evaluating type II resistance [21]. Based on SFI, a BBFI method has been developed to further improve the successful infection rate in field conditions without moistening facility [30]. BRII is a relatively novel method that we have recently reported, and is used in evaluating type II resistance [25]. One of the prominent merits of BRII method lies in exemption of moisture-maintaining system when compared with other existing methods [25]. UBFI method was developed here to avoid PSD that frequently occurred in susceptible FHB-genotypes by SFI and BBFI inoculation methods (Figure 1 and Table 1).

PSD is a common phenomenon in the outbreak of FHB, which affects the normal grain filling process of the whole spike [34]. In this study, we defined the ratio of the number of prematurely dead spikes to the total inoculated wheat spikes under a certain inoculation method as proportion of premature spike death (PPSD). UBFI, BBFI, and BRII had different effects on PPSD. Under the UBFI method, the pathogen mainly spread downward and had little influence on the growth and development of the pathogen-unreached spikelets below the inoculation point, so the bleaching of spikes went progressively downward instead of causing acute premature death, i.e., PSD. Additionally, the UBFI method resulted in a negligible difference in disease severity between the two contrasting groups of *Fhb1* vs. non-*Fhb1* within 14 days after infection, and the difference in disease severity between the contrasting groups of *Fhb1* vs. non-*Fhb1* became remarkable at a later stage of infection (Figure 2A–C). In a word, the UBFI method seemed to reflect the nature of resistance to disease spread within the spike without causing acute PSD. In contrast, BBFI frequently caused acute death of the whole wheat spike in susceptible genotypes (Appendix A). In this case, the pathogen entering the rachis damaged the vascular tissue of the spike, and the spikelets above the infection point eventually starved to death due to the lack of nutrient and water supply. The probability of PSD in the lines without *Fhb1* under BBFI was close to 100% (Figure 4 and Table 1), so BBFI maximized the phenotypic differences between resistant and susceptible genotypes (Figure 2A–C). In this case, attributed to the hemi-biotrophic lifestyle of *Fusarium graminearum* [35], it is suspicious to define the acute spikelets death as type II resistance since the pathogen may not spread within the dead spike due to the lack of humidity and nutrients required for pathogen growth.

For the BRII method, the pathogen inoculum was injected into the basal rachis internode, and the spikelets could be infected only when the pathogen passed through the rachis node, which demonstrated that the rachis node was crucial for hindering the spread of pathogen from one spikelet to another. The majority of lines carrying *Fhb1* had no visible diseased spikelets at 21 DAI (Table 1 and Appendix A), which confirmed our previous observation [25]. The lines without *Fhb1* also had a high PPSD but lower than that of BBFI (Figure 4 and Table 1). PPSD under the BBFI and BRII methods were significantly different between the NILs with contrasting *Fhb1* alleles (*p* < 0.01) (Figure 4), indicating that PPSD, instead of PSS, could be an optimal and practical index of FHB severity and could be used to reflect the genetic effect of *Fhb1* on disease severity.

The robustness and accuracy of phenotypic evaluation is the prerequisite for fine mapping, positional cloning, functional characterization, and mechanism understanding. The mixture of PSS with PPSD brought to mind the controversial issues of the candidate genes, conflicting functions, and mysterious mechanism of *Fhb1* [36,37,38]. In addition to phenotypic issue, other factors including the huge and complex wheat genome, the complexity of pathogen–wheat interaction and sensitivity to environmental cues might also contribute to the difficulty of unveiling the mysterious of *Fhb1*.

### 3.2. UBFI Was Suitable for Assessing Mycotoxin Accumulation Potential

Mycotoxins, such as DON, NIV, etc., are virulence factors in the process of pathogen-host interaction, which are not only one of the factors that aggravate the severity of FHB, but are also harmful to the health of consumers [7]. Therefore, it is necessary to assess the degree of toxin contamination in wheat grains and to develop varieties with low mycotoxin accumulation potential. Until now, the genetic research on type III resistance has lagged behind type II resistance due to a lack of appropriate inoculation methods, high cost of assay, large within-genotype variations, virulence of pathogen, etc., which obviously requires more effort and time. Most QTL for type III coincide with those for type II [17]. The wheat spikes without *Fhb1* were more seriously contaminated by DON than the spikes with *Fhb1*, but several lines carrying *Fhb1* have also been reported to have extremely high DON content [30]. A transcript mapping study showed that the DON-responsive transcripts were associated with, but not exclusively located within, *Fhb1* [31]. In practice, the visual rating finishes 3–4 weeks after inoculation, while mycotoxin analysis is carried out after wheat harvest [23], and various uncertain factors during this additional time have a high probability to affect the evaluation of type III resistance.

A suitable inoculation method that significantly distinguishes different wheat genotypes in mycotoxin content is essential for research on type III resistance. At present, SFI and spray/grain-spawn inoculation, as common inoculation methods for assessing FHB severity, have been used directly in mycotoxin analysis [30,39,40], whereas little was known about the effects of different inoculation methods on mycotoxin accumulation. Unexpectedly, the BBFI method did not distinguish DON content between the NILs differing in *Fhb1* in this study (Figure 2D–F). The current data clearly demonstrated that DON content was highly significantly different (*p* < 0.01) between resistant and susceptible genotypes under the UBFI method, and significantly different (*p* < 0.05) under the BRII method (Figure 2D–F), which indicated that the effect of *Fhb1* on mycotoxin accumulation depended on inoculation methods, and also might explain why DON content had been highly variable among plants within a genotype/different replications under spray/spawn methods due to random infection points.

The mechanism by which the infection point of pathogenic fungi affects the mycotoxin accumulation in grains is not easy to explain. Mycotoxins are water-soluble and, theoretically, can be transported up and down the vascular tissue of spike and stem. UBFI and BBFI were performed by injecting inoculum into spikelets at the top and bottom of the spike, respectively. One possible explanation for the phenomenon of high DON content under UBFI method is that no PSD occurs and the pathogen progresses chronically downward; thus, the pathogen-unreached spikelets stay alive and sustain the supply of nutrients that induce the pathogen to produce DON. Another possible explanation may be that DON tends to be transported downwards, so DON may have a greater probability of remaining in the healthy grains. Our previous studies in a population of recombinant lines derived from Ning7840 and Clark showed that the effects of *Fhb1* on DON content between the BBFI and BRII methods were similar [30]. The discrepancy between the previous and current experiments may be attributed to the genetic backgrounds. DON content and disease severity are quantitative traits. In current study, the genetic backgrounds of the NILs were of recurrent parent Clark, and the foreground was determined mainly by *Fhb1* locus.

### 3.3. The Relationship between PSS and DON Content Was Subjected to Inoculation Methods

The relationship between FHB severity and toxin accumulation has been somewhat complex and conflicting. Varying degrees of relationship between FHB severity and DON content have been reported, including high positive correlations, low positive correlations, and no correlations [28,41,42]. A meta-analysis across 163 studies supported the high correlation between FHB severity and DON content (*p* < 0.001) [27]. As discussed above, we thought that BBFI and UBFI were beneficial for assessment of FHB severity and DON accumulation, respectively, and, in this case, the positive and significant correlation (*p* < 0.05) between the two traits validated most of the previous studies (Figure 3). In addition, no correlation (*p* > 0.05) between FHB severity and DON content in grains was found under the BBFI method, whereas the correlation was weakly significant under the BRII method (*p* < 0.05, *r* ≤ 0.4) (Figure 3). Therefore, the degree of relationship between the two traits was greatly affected by inoculation methods or infection points. The visual FHB rating has become a useful parameter for DON prediction [28], but large variations in DON levels would be expected in the case of random infection points or the occurrence of PSD. If the infection point occurs in the upper part of a spike, the wheat grains may be contaminated by more mycotoxins, while the whole spike with severe premature death may accumulate less toxin.

## 4. Conclusions

In this study, the near-isogenic lines contrasting in *Fhb1* alleles were used as materials to analyze the advantages, disadvantages, and applicability of the three phenotyping methods for disease severity and DON accumulation in grains. We concluded no inoculation method was versatile, and infection point (or inoculation method) was one of the important factors contributing to the complicated and uncertain relationship between disease severity and mycotoxin accumulation. UBFI narrowed the difference in PSS between resistant and susceptible lines, which might not be competent for evaluating disease severity for breeders. However, UBFI had prominent advantages over other methods for DON assay. BBFI was an optimized method from SFI, which maximized phenotypic differences between resistant and susceptible genotypes of FHB, and was suitable for evaluating the disease severity of a genotype, but would not be recommended for evaluation of mycotoxin accumulation potential. BRII was complementary to BBFI and UBFI. PPSD can be a key index to evaluate wheat resistance to FHB. Therefore, it is critical to select an appropriate phenotyping method or a combination of two or more methods to develop wheat elite cultivars with overall resistance to FHB.

## 5. Materials and Methods

### 5.1. Plant Materials

Twenty-two wheat near-isogenic lines (NILs) contrasting in *Fhb1* alleles were used in this study. The NILs for *Fhb1* were generated from the heterozygotes at *Fhb1* locus derived from the cross of R75 and S98, two advanced backcrossed lines contrasting in *Fhb1* alleles, which were previously developed from the cross of Ning7840 and Clark by Dr. Guihua Bai at Kansas State University. R75 carries *Fhb1* and S98 does not. The experiments were conducted in the experimental field of Yangzhou University (119°42′ E, 32°39′ N), Jiangsu, P. R. China in two consecutive years, 2020 and 2021. Each line was planted in 6 rows, with 15 plants in each row, and 2 replicates per line each year.

### 5.2. Fusarium graminearum Strain and Inoculum Preparation

A *Fusarium graminearum* strain Fg0865 (15-ADON chemotype) with strong pathogenicity and high sporulation was used for inoculation [25]. The strain was provided by Professor Huaigu Chen in Jiangsu Academy of Agricultural Sciences. The strain was activated by culturing on potato dextrose agar (PDA) medium at 25 °C for 4–5 days. The blocks of activated strain were then taken with a puncher with a diameter of seven millimeters (mm) and cultured in the mung bean soup with a volume of 10 milliliters (mL) per block at 25 °C and 150 r/min for 3–5 days to induce sporulation. The spore density was determined by a hemocytometer under the microscope. The inoculum reached at least 10^5^ spores per microliter (μL) for UBFI and BBFI methods, and 10^6^ spores/μL for BRII inoculation method.

### 5.3. FHB Inoculation and Phenotypic Evaluation

At the early flowering stage of wheat, three inoculation methods of UBFI, BBFI, and BRII were used for FHB inoculation. UFBI and BBFI methods were performed by injecting about 10 μL inoculum into the bilateral florets of a spikelet positioned at the fifth spikelet in the upper part of a spike and at the fifth spikelet from the bottom of a spike, respectively. The BRII inoculation was performed by injecting about 1 μL concentrated inoculum into the basal rachis internode of a spike. At least 15 spikes were inoculated for each line per replicate under each method.

At the seventh day after inoculation (DAI), the wheat spikes without visible symptoms at the inoculated site were removed due to inoculation failure. The proportion of symptomatic spikelets (PSS) per spike was counted at 21 DAI, which was calculated as “number of symptomatic spikelets/total number of spikelets of a spike”. The proportion of premature spike death (PPSD) under BBFI and BRII methods was calculated as “number of prematurely dead spikes/total number of inoculated spikes”.

### 5.4. Extraction and Quantification of DON in Grains

At maturity, all the inoculated spikes of each line were harvested and threshed manually to retain FHB-infected grains, and then ground into powder. DON in a gram of powder sample was extracted with 4 mL mixed solution of 49.5% acetonitrile,1% formic acid, and 49.5% distilled water. The extract was filtered sequentially through pore diameters of 0.45 μm and 0.22 μm needle-type organic phase filters into the autosampler insert (8 mm, 200 μL) of the autosampler vial (8 mm, 2 mL). All prepared samples were stored at −20 °C for subsequent determination. DON content in grains was determined following the protocol as described by Mao et al. [43], using liquid chromatography-triple quadrupole mass spectrometry (LC-MS/MS, TSQ-Vantage, Thermo Fisher SCIENTIFIC, Waltham, MA, USA).

### 5.5. Data Analysis

Statistical analysis was performed using Excel 2016 (Microsoft Office Inc., Redmond, DC, USA) and IBM SPSS Statistics 21.0 (IBM, Armonk, NY, USA). Moreover, best linear unbiased prediction (BLUP) was also performed to remove environmental effects in an R package called lme4 operated by RStudio 1.4 software (Rstudio, Boston, MA, USA). The effects of *Fhb1* alleles and interannual variation on PSS, DON content and PPSD under the three inoculation methods were determined by one-way ANOVA and *t* test, respectively, and the linear correlations among PSS, DON content, and PPSD under the three methods were measured using correlation analysis set from the simple Pearson correlation coefficient, using significance levels of 0.05 and 0.01. OriginPro 2021 (OriginLab, Northampton, MA, USA) and Adobe Photoshop CC 2019 (Adobe System Incorporated, Mountain View, CA, USA) were used for graphic processing.

## Figures and Tables

**Figure 1 toxins-14-00409-f001:**
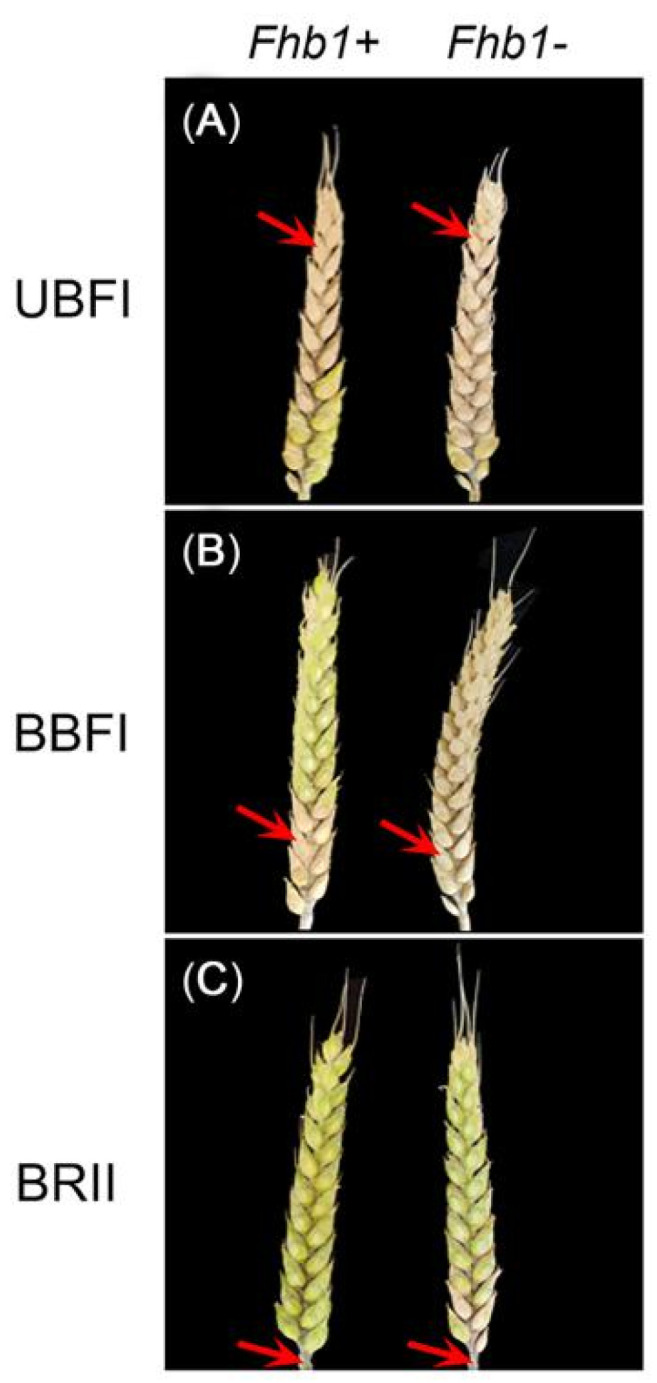
FHB severity of the *Fhb1* near-isogenic lines at 21st day after inoculation under the three inoculation methods. (**A**–**C**) Contrasting FHB symptoms of *Fhb1* NILs under the UBFI, BBFI, and BRII methods, respectively. The wheat spike with *Fhb1* on the left showed a significantly lower level of FHB severity than the corresponding spike without *Fhb1* on the right under each method. The red arrows point to the inoculation sites. The inoculum was injected into the bilateral florets of a spikelet under the UFBI and BBFI methods, and BRII inoculation was performed by injecting into the basal rachis internode.

**Figure 2 toxins-14-00409-f002:**
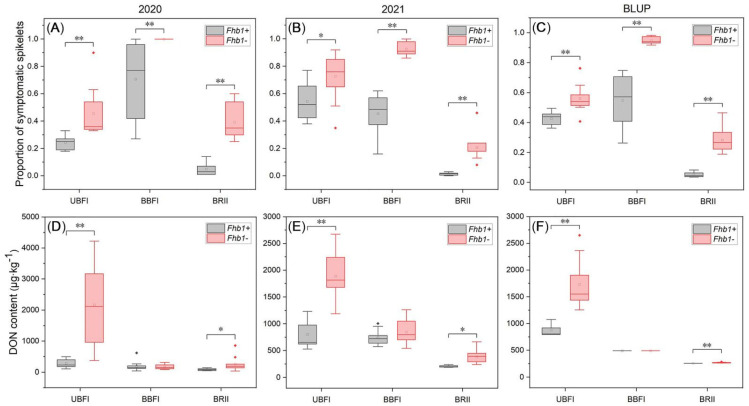
Distribution of proportion of symptomatic spikelets (PSS) and deoxynivalenol (DON) content in the *Fhb1* near-isogenic lines under the three phenotyping methods. (**A**,**D**) Year 2020. (**B**,**E**) Year 2021. (**C**,**F**) Predicted PSS and DON content by best linear unbiased prediction (BLUP). The statistically significant differences between groups are labeled with ‘*’ at *p* < 0.05 and ‘**’ at *p* < 0.01 (Fisher’s least significant difference, LSD). The box ends indicate the upper (3rd) to lower (1st) quartiles of the value ranges, and the whiskers indicate the highest and the lowest values. The horizontal line inside the box marks the median for the phenotypic values. The hollow block inside the box marks the mean for the phenotypic values. The filled diamond outside the box marks the outliers for the phenotypic values.

**Figure 3 toxins-14-00409-f003:**
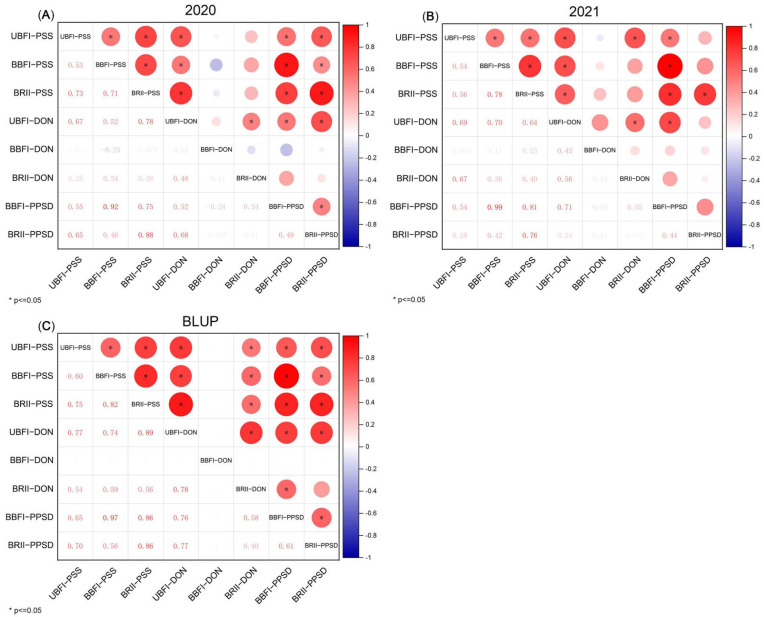
The Pearson’s correlation coefficients for proportion of symptomatic spikelets (PSS), deoxynivalenol (DON) content and proportion of premature spike death (PPSD). (**A**) Year 2020. (**B**) Year 2021. (**C**) Predicted PSS, DON content, and PPSD by best linear unbiased prediction (BLUP). The red and blue circles on the upper right-side indicate significant positive and negative correlations, respectively, and are marked inside with ‘*’ at *p* < 0.05, and the empty cases refer to insignificant correlations. The color gradient is proportional to the correlation coefficient. The correlation coefficients are marked on the corresponding lower left-side.

**Figure 4 toxins-14-00409-f004:**
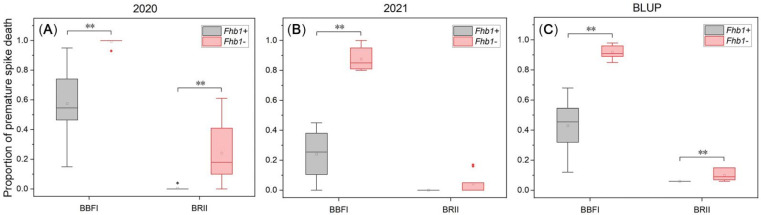
Distribution of proportion of premature spike death (PPSD) in the *Fhb1* near-isogenic lines under the methods of basal bilateral floret injection (BBFI) and basal rachis internode injection (BRII). (**A**) Year 2020. (**B**) Year 2021. (**C**) Predicted PPSD by best linear unbiased prediction (BLUP). The statistically significant differences between groups are labeled with ‘**’ at *p* < 0.01 (Fisher’s least significant difference, LSD). The box ends indicate the upper (3rd) to lower (1st) quartiles of the value ranges, and the whiskers indicate the highest and the lowest values. The horizontal line inside the box marks the median for the phenotypic values. The hollow block inside the box marks the mean for the phenotypic values. The filled diamond outside the box marks the outliers for the phenotypic values.

**Table 1 toxins-14-00409-t001:** Differences in PSS, DON content in grains, and PPSD in the near-isogenic lines contrasting in *Fhb1* alleles.

		PSS			DON Content in Grains (μg∙kg^−1^)			PPSD		
Inoculation Method	Year	*Fhb1^−^*	*Fhb1^+^*	*Diff.* & *Sig.*	*Fhb1^−^*	*Fhb1^+^*	*Diff.* & *Sig.*	*Fhb1^−^*	*Fhb1^+^*	*Diff*. & *Sig.*
UBFI	2020	0.46 ± 0.18	0.24 ± 0.05	0.22 ± 0.23 **	2168.78 ± 1345.76	274.73 ± 126.20	1894.05 ± 1471.96 **	—		
	2021	0.73 ± 0.17	0.55 ± 0.15	0.18 ± 0.22 *	1890.31 ± 435.95	803.11 ± 293.95	1087.20 ± 729.90 **			
	BLUP	0.56 ± 0.09	0.43 ± 0.04	0.13 ± 0.13 **	1733.55 ± 453.56	878.58 ± 122.04	854.97 ± 575.60 **			
BBFI	2020	1.00 ± 0.00	0.71 ± 0.28	0.29 ± 0.28 **	165.28 ± 71.36	186.51 ± 155.26	−21.24 ± 226.62	0.99 ± 0.02	0.58 ± 0.24	0.41 ± 0.26 **
	2021	0.92 ± 0.05	0.45 ± 0.16	0.47 ± 0.21 **	848.63 ± 214.73	743.38 ± 141.17	105.25 ± 355.89	0.88 ± 0.08	0.24 ± 0.17	0.64 ± 0.25 **
	BLUP	0.95 ± 0.02	0.55 ± 0.17	0.40 ± 0.19 **	490.71 ± 0.00	490.71 ± 0.00	0.00	0.92 ± 0.04	0.43 ± 0.18	0.49 ± 0.22 **
BRII	2020	0.39 ± 0.12	0.05 ± 0.04	0.34 ± 0.16 **	261.12 ± 226.62	86.08 ± 31.02	175.04 ± 257.64 *	0.24 ± 0.21	0.01 ± 0.01	0.23 ± 0.22 **
	2021	0.21 ± 0.10	0.02 ± 0.01	0.19 ± 0.11 **	382.05 ± 121.01	204.86 ± 22.98	177.19 ± 143.99 *	0.04 ± 0.06	0.00 ± 0.00	0.04 ± 0.06
	BLUP	0.28 ± 0.08	0.05 ± 0.02	0.23 ± 0.10 **	268.01 ± 6.82	257.17 ± 0.45	10.84 ± 7.27 **	0.10 ± 0.03	0.06 ± 0.00	0.04 ± 0.03 **

PSS—proportion of symptomatic spikelets; DON—deoxynivalenol; PPSD—proportion of premature spike death; UBFI—upper bilateral floret injection; BBFI—basal bilateral floret injection; BRII—basal rachis internode injection; *Fhb1^−^*—the lines without *Fhb1*; *Fhb1^+^*—the lines carrying *Fhb1*; BLUP— best linear unbiased prediction. The statistically significant differences between groups are labeled with ‘*’ at *p* < 0.05 and ‘**’ at *p* < 0.01 (Fisher’s least significant difference, LSD).

## Data Availability

The original contributions presented in the study are publicly available.

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
