# Peer review of "Mycotoxin DON Accumulation in Wheat Grains Caused by Fusarium Head Blight Are Significantly Subjected to Inoculation Methods"

_toxins, 2022, doi:10.3390/toxins14060409_

Round 1

Reviewer 1 Report

Mycotoxin deoxynivalenol (DON) contamination of wheat and other grains has been a concern of food safety. Minimizing DON contamination in grains is a major goal of Fusarium Head Blight (FHB) research community. While DON concentrations in grains are usually correlated with disease severity, high DON levels have been observed in asymptomatic kernels. Studies show that infection timing has effect on disease severity and DON distribution in wheat spikes, and a late infection is an important factor for high DON levels found in healthy-looking kernels. Authors in this manuscript report that DON accumulation in wheat grain is affected by inoculation methods (infection points). Among the three methods used, the upper bilateral floret injection (UBFI) maximizes DON accumulation in grain and significantly distinguishes DON content in different wheat genotypes, indicating that the method could be useful for breeding program to develop wheat cultivars with resistance to DON accumulation. I would like to recommend its acceptance for publication. The following are some comments and suggestions.

1.     I would like to suggest authors label inoculation points in Figure 1, which could give readers a clear view of three different inoculation methods.

2.     Line 142: “proportion of PPSD” should be “proportion of PSD or PPSD”.

3.     Line 143: “PPSD-premature spike death” should be “PPSD-proportion of premature spike death”.

4.     Lines 168 - 169: Should it be that DON content of most lines without Fhb1 exceeded 1000 µg/kg under UBFI? Where is 4224.38 µg/kg in Table 1? Do you mean in Table S1?

5.     Lines 411 & 413: “PPSB” should be “PPSD”.

6.     Table S1: “PPSD - premature spike death” in the footnote should be “PPSD - proportion of premature spike death”.

7.     Fhb1- and Fhb1+ don’t appear in Table S1, so they can be removed from the footnote.

Reviewer 2 Report

The paper describes a set of experiments carried out in 2020 2021 on isogenic lines of wheat using different methods of Fusarium inoculation in order to assess the effect of Fusarium inoculation on FHB disease index and DON accumulation.

The paper is of high interest for the community working on FHB and shows the importance of inoculation method on the results of disease and mycotoxin accumulation. It is clearly written so it is suitable for publication given some minor changes/modifications:

    1)       I would suggest to avoid citing supplementary figures at the beginning of the chapter.

See for example chapter 2.4. Given that 2.4 chapter discusses data of supplementary figure 4 I would suggest to make figure S4 as Figure 4 of the manuscript.

     2)       To futher clarify the infection methods a supplementary figure detailing graphically the modes of inoculation with figures would make the paper extremely useful for the community

     3)       Identity of the strain used for infection should be confirmed: a multilocus species characterisation is needed to confirm the species of the strain or a reference to a publication where the strain was described and appropriately characterised.

     4)       Mycotoxin data analysis should be provided (I guess is policy of mdpi to make raw data available together with the publication)
